# WATCH AND LEARN: LEARNING TO USE COMPUTERS FROM ONLINE VIDEOS

## ABSTRACT

Computer use agents (CUAs) need to plan task workflows grounded in diverse, ever-changing applications and environments, but learning is hindered by the scarcity of large-scale, high-quality training data in the target application. Existing datasets are domain-specific, static, and costly to annotate, while current synthetic data generation methods often yield simplistic or misaligned task demonstrations. To address these limitations, we introduce *Watch & Learn* (*W&L*), a framework that converts human demonstration videos readily available on the Internet into executable UI trajectories at scale. Instead of directly generating trajectories or relying on ad hoc reasoning heuristics, we cast the problem as an inverse dynamics objective: predicting the user's action from consecutive screen states. This formulation reduces manual engineering, is easier to learn, and generalizes more robustly across applications. Concretely, we develop an inverse dynamics labeling pipeline with task-aware video retrieval, generate over 53k high-quality trajectories from raw web videos, and demonstrate that these trajectories improve CUAs both as in-context demonstrations and as supervised training data. On the challenging OSWorld benchmark, UI trajectories extracted with W&L consistently enhance both general-purpose and state-of-the-art frameworks in-context, and deliver stronger gains for open-source models under supervised training. These results highlight web-scale human demonstration videos as a practical and scalable foundation for advancing CUAs towards real-world deployment.

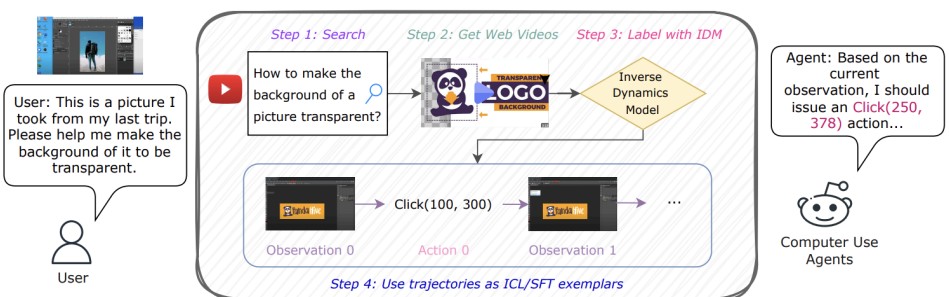

Figure 1: W&L converts web-scale human demonstration videos into executable UI trajectories, providing scalable supervision and in-context exemplars for computer use agents.

## 1 INTRODUCTION

Computer use agents (CUAs) Zheng et al. (2024a); Deng et al. (2023); Qin et al. (2025); Gou et al. (2025); OpenAI (2025b) hold the promise of transforming how humans interact with software and the web, from everyday productivity tasks to enterprise-scale automation. To be effective, CUAs must both *plan* multi-step task workflows that incorporate domain knowledge, and *ground* these plans into concrete UI actions within diverse and ever-changing applications. Progress toward these capabilities hinges on access to high-quality task demonstrations, yet collecting annotated trajectories at scale is prohibitively expensive.

Meanwhile, the web is rich in human demonstration videos (e.g., YouTube tutorials, screencasts, etc.), which naturally encode complex workflows across diverse applications. Unlocking this resource could provide CUAs with scalable supervision and rich priors for expert-level planning. However, existing synthetic data generation approaches have fallen short of realizing this vision.

Prior efforts fall into three main categories: *Offline synthesis* attempts to recover trajectories from videos using pipelines that combine multimodal large language models (MLLMs) with UI element detectors and transition parsers. Despite substantial engineering, systems such as MONDAY (Jang et al., 2025b) and TongUI (Zhang et al., 2025) achieve only modest action labeling accuracies (∼70% for MONDAY), reflecting the limitations of multi-stage heuristics. *Online synthesis* generates trajectories through random exploration in real-world environments and later retrofits them with pertinent task instructions (Murty et al., 2024; Sun et al., 2025). While scalable in principle, this approach produces low-complexity demonstrations that are less aligned with human goals and can be costly as they require online exploration. *Hybrid approaches*, such as Explorer (Pahuja et al., 2025), generate task proposals and then execute and refine them online, but still rely on MLLMs for action grounding—thereby sharing similar limitations to offline synthesis methods.

Overall, these approaches either rely on brittle heuristics, are costly as they rely on explorations in real environments, or generate low-complexity demonstrations misaligned with human intent. To address these limitations, this work introduces **Watch & Learn (W&L)**, a framework that converts human demonstration videos readily available online into executable UI trajectories at scale (Figure 1). Instead of directly generating trajectories or depending on complex multi-stage pipelines, we frame the problem as an *inverse dynamics* objective: given two consecutive observations $(O_t, O_{t+1})$, predict the intermediate action $a_t$ that produced the transition. This formulation is easier to learn, avoids hand-crafted heuristics, and generalizes robustly across applications. In robotics, inverse dynamics modeling is a well-established method for recovering actions from state transitions (e.g., VPT (Baker et al., 2022), DreamGen (Jang et al., 2025a)); here, we demonstrate that the same principle can be adapted effectively for CUAs. From our experiments, this simple formulation yields a highly accurate model of user behavior, sidestepping the complexity of conventional pipelines.

To scale this approach to the web, we construct a large state-transition corpus of 500k state transition data from real-world web interactions. Each example consists of an observation at time $t$, an action, and the resulting observation at $t + 1$. Training an inverse dynamics model (IDM) on this corpus allows us to directly map visual transitions into structured actions. We further design a retrieval framework that retrieves YouTube videos relevant to target tasks (for in-context learning) or general video tutorials (for supervised fine-tuning). Applying the IDM to these videos transforms raw demonstrations into high-quality trajectories, covering a broad spectrum of real-world workflows.

Beyond data collection, W&L uncovers a different role for CUAs. In addition to effectively using UI trajectories in training, we demonstrate that the extracted trajectories can also serve as *in-context exemplars* during inference, enabling CUAs to leverage planning and grounding priors enriched with domain knowledge on the fly. This dual role (training and in-context guidance) enables flexible integration with both open-source models and general-purpose agents. To illustrate the effectiveness of this approach, we evaluate W&L on OSWorld (Xie et al., 2024), a challenging benchmark requiring both domain familiarity and strong planning and grounding capabilities. On OSWorld, trajectories extracted from web-scale videos deliver consistent gains: in-context use improves general-purpose models and state-of-the-art agentic frameworks by up to 3 percentage points, while training with them yields even larger improvements for open-weight models (up to 11 percentage points). Importantly, these benefits are achieved without any manual annotation, demonstrating that web-scale human workflows can serve as a practical and scalable foundation for advancing CUAs towards real-world deployment.

In summary, our contributions are three-fold: *(i)* We develop a scalable inverse dynamics labeling pipeline, coupled with a task-aware video retrieval framework, that transforms raw web videos into high-quality trajectories. Overall, without any manual effort, we generate 53,125 trajectories with high-accuracy action labels. *(ii)* We show that these video-derived trajectories can serve as *in-context demonstrations* at inference time, improving general-purpose CUAs without retraining. *(iii)* We also demonstrate that these trajectories provide effective *training data*, offering a scalable supervision signal that substantially improves open-source CUAs.

## 2 RELATED WORK

### 2.1 DATA SYNTHESIS FOR COMPUTER USE AGENTS

While human-curated UI control datasets have been collected (Deng et al., 2023; Lù et al., 2024; Rawles et al., 2023; Li et al., 2024), their limited size and diversity remains a key bottleneck for CUAs. Recent work has focused on synthesizing data from exploration, tutorials, or self-play.

Exploration-based approaches such as BAGEL (Murty et al., 2024), NNetNav (Murty et al., 2025), Explorer (Pahuja et al., 2025), and OS-Genesis (Sun et al., 2025) generate training data by letting agents explore websites and retroactively labeling their interactions with task instructions. This paradigm yields scalable but often noisy data, with alignment and accuracy depending heavily on heuristics or MLLM labeling. Other methods leverage online resources: Synatra (Ou et al., 2024) and AgentTrek (Xu et al., 2025) transform textual tutorials into executable trajectories, while TongUI (Zhang et al., 2025) aggregates a massive corpus of multimodal tutorials (text and screen-cast videos) into GUI interaction data. These approaches demonstrate that web-scale instructional content can provide diverse coverage across applications, but they rely primarily on off-the-shelf MLLMs to label trajectories, which often introduces brittleness or misalignment.

Another line of work integrates synthesis into the training loop itself. OpenWebVoyager (He et al., 2025) improves through online exploration and feedback; WebRL (Qi et al., 2025) generates new instructions from failed tasks to form a self-evolving curriculum; SCA (Qi et al., 2025) has agents self-generate and verify new tasks in a code-as-task format; and ZeroGUI (Yang et al., 2025) proposes a fully automated online learning framework for GUI agents, where VLMs generate tasks and rewards that drive reinforcement learning without manual annotations. These strategies enable continual improvement without additional human data, but often produce simplistic or narrow task distributions. Moreover, the process can be expensive as it involves multiple iterations of data generation and training.

Our framework, *Watch & Learn*, also leverages web videos like TongUI (Zhang et al., 2025), but differs in its technical strategy. Instead of relying on MLLMs to label tutorial steps, we train an inverse dynamics model (IDM) that can accurately infer user actions from consecutive screen states. This produces highly reliable UI trajectories that not only provide stronger supervised training signals but also serve as more effective in-context exemplars at inference time. By combining web-scale video mining with accurate action labeling, our approach complements prior work and highlights the value of extracting accurate cues from video-based supervision for CUAs.

### 2.2 IN-CONTEXT LEARNING FOR AGENTS

In-Context Learning (ICL) has emerged as a pivotal test-time scaling paradigm for large language models, enabling them to adapt to new tasks without explicit parameter updates (Dong et al., 2022). This approach is particularly useful for enhancing LLM-powered agentic systems (Su et al., 2025).

Despite being generally helpful, the effectiveness of ICL is heavily influenced by the scale of the LLMs and the size of their context window, particularly for long-horizon, multi-step tasks. While including more ICL examples usually brings performance gains (Agarwal et al., 2024), this method incurs significant computational overhead and latency with long demonstration trajectories. Therefore, efficiently selecting demonstration sequences (Gupta et al., 2025) or abstracting them in high-level workflows (Wang et al., 2024; Zheng et al., 2024b) has become a promising research direction. For computer-use agents, where tasks are often long and complex, one major challenge is the model's inability to plan effectively. Several pieces of work have leveraged ICL to address this specific problem (Holt et al., 2025; Zhao et al., 2025).

Another important direction is to develop data-centric frameworks to adapt LLM agents to any given environments without human annotations (Su et al., 2025). However, such methods require generating large amounts of synthetic data, and the potential for using publicly available web-scale video data as ICL examples still remains underexplored.

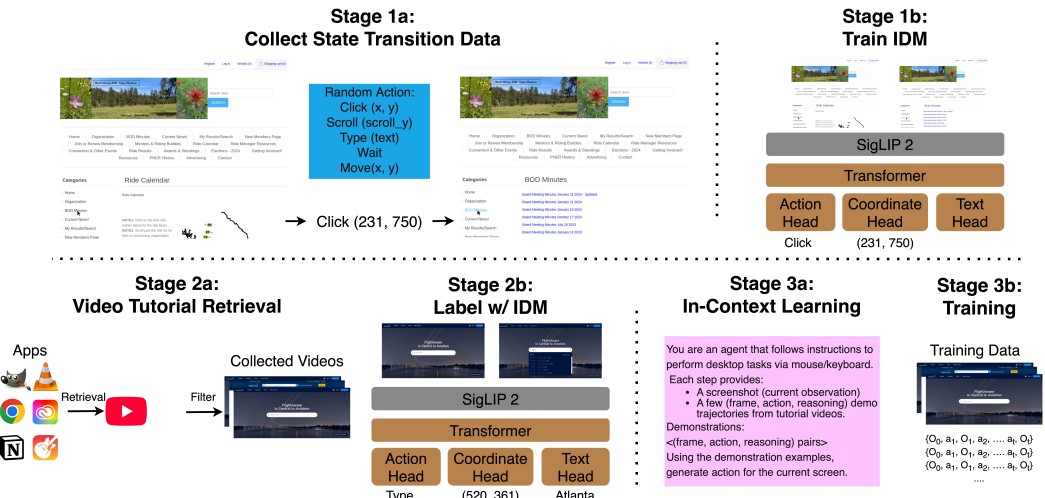

Figure 2: **Method overview.** Our framework converts web-scale human demonstration videos into executable trajectories for CUAs. We first collect a large-scale state-transition dataset of screen observations and user actions, and train an inverse dynamics model (IDM) to recover actions from consecutive screenshots. This IDM is then applied to tutorial videos to extract step-by-step trajectories. A retrieval module selects task-relevant or general demonstrations, which are used in two ways: (i) as in-context exemplars that provide application-specific knowledge at inference time, and (ii) as supervised training data to improve open-source CUAs.

## 3 METHOD

Computer use agents must operate the user interface of many diverse and ever-changing applications where internal UI representations such as HTML or accessibility trees are often incomplete, inconsistent, or unavailable. To maximize generality and scalability, we focus on a *vision-only* setting: models observe raw screen pixels and output structured user actions. This mirrors how humans interact with computers, by visually perceiving the interface and deciding where to click or what to type, while avoiding brittle dependencies on application-specific APIs or noisy UI representations.

At a high level, our framework works in three stages (see Figure 2). First, we construct a large-scale state-transition corpus from diverse computer interaction data and use it to train an inverse dynamics model (IDM), enabling the system to recover the underlying actions from consecutive screen observations. Second, we apply this IDM to web-scale tutorial videos, paired with a retrieval component that identifies either task-relevant videos (for inference-time use) or general tutorials (for training). This process automatically produces executable UI trajectories without manual labeling. Finally, we leverage these trajectories in two complementary ways: as *in-context exemplars*, which provide CUAs with planning and grounding priors as well as application-specific knowledge at inference time; and as *supervised training data*, which can be used to fine-tune models and improve their general knowledge.

### 3.1 INVERSE DYNAMICS MODEL

A key component of our framework is an IDM that predicts the user action given two consecutive screen observations. Training such a model requires large-scale state-transition data, which is scarce in existing datasets. To address this gap, we construct our own corpus of transitions by synthesizing interactions at scale, complemented by existing human-collected datasets.

**State-transition data collection.** To obtain large-scale supervision, we built an automated data generation pipeline that interacts with live web pages and records state transitions. Inspired by WebDreamer (Gu et al., 2025), we randomly select entry points from the March 2025 Common Crawl index and launch browsing sessions that perform sequences of actions such as clicking, typing

text, scrolling, and moving the cursor. The action policy is not uniform: we weight the sampling toward common interactions (e.g., clicks) while still ensuring that less frequent actions are covered. Through this procedure, we collected around 500k synthetic transitions. To complement these, we also incorporate 132k human-annotated transitions from the Mind2Web dataset (Deng et al., 2023), yielding a training corpus of more than 630k $(O_t, a_t, O_{t+1})$ triples.

**Model architecture.** The IDM takes as input two consecutive observations $(O_t, O_{t+1})$ and outputs the action $a_t$ that caused the transition. We adopt a vision-only architecture consisting of a SigLIP-2 vision encoder followed by four Transformer Vaswani et al. (2017) layers. On top of this backbone, we attach three specialized prediction heads:

- **Action classification head:** a categorical predictor over five supported primitives: `click`, `scroll`, `type`, `wait`, and `move`.
- **Coordinate head:** for location-based actions (click, move, type), the model predicts normalized $(x, y)$ coordinates discretized into integers from 0 to 1000. This converts coordinate regression into a classification problem, which proved to be more stable in training.
- **Language head:** for text entry actions, the model generates the string input using a GPT-2 small decoder (Radford et al., 2019) attached to the Transformer backbone.

Scroll and wait actions require no additional arguments; the model simply predicts their occurrence.

**Training and evaluation.** The IDM is trained with a multi-task objective: cross-entropy for action class prediction, cross-entropy for discretized coordinates, and language modeling loss for text generation. Training is performed end-to-end over the 630k transition corpus. We evaluate the IDM on the held-out test split of Mind2Web (Deng et al., 2023), which provides human-annotated trajectories across diverse websites. This benchmark allows us to measure both action classification accuracy and argument prediction quality in a realistic setting. As reported in Section 4.2.2, our IDM trained on state transition data achieves stronger action accuracy than off-the-shelf foundation models, validating its effectiveness as the core labeling module in our framework.

### 3.2 Data Generation from Videos

Once the IDM is trained, we retrieve suitable tutorial videos and apply the IDM.

**Video retrieval.** We build a retrieval framework that searches and downloads tutorial videos from large video platforms such as YouTube. The retrieval procedure differs depending on whether the goal is inference-time support or large-scale training data collection. *Inference-time retrieval.* Given a task description and the target application, we form a natural language search query. To refine the query, we prompt Gemini 2.5 Flash[1] (Gemini Team, 2025) with both the task instruction and the initial screen, asking it to generate a more specific query. We then use the YouTube Search API to retrieve the top 15 videos. For example, a task instruction `"Can you increase the max volume of the video to the 200% of the original volume in VLC?"` becomes the search query `"vlc increase max volume"`. Each retrieved video is paired with its title, which we treat as the candidate task description. *Training-time retrieval.* To construct a broad training dataset, we curate a list of 69 applications spanning productivity, programming, design, screen editing, audio production, system utilities, and science/data domains. For each one, we prompt Gemini 2.5 Flash to generate plausible task queries and use them to search on video platforms, downloading the corresponding tutorial videos.

**Filtering.** Not all retrieved videos are usable. We sample frames at 1 frame per second and automatically filter out segments that are not screencasts (e.g., talking-head segments), are zoomed in/out, or are blurred due to transitions. Gemini 2.5 Flash is used as a classifier to perform this filtering. For inference-time retrieval, we retain only the top 3 videos that pass filtering to minimize noise. For training data collection, we keep all videos that satisfy the filter.

**Trajectory labeling.** After filtering, we segment each video into a sequence of frames $\{O_0, O_1, \ldots\}$ and apply the IDM to every consecutive pair $(O_t, O_{t+1})$, predicting the intermediate action $a_t$ and assembling a trajectory $\tau = (O_0, a_0, O_1, a_1, \ldots, O_T, a_T, O_{T+1})$. In this way, raw human demonstration videos are transformed into structured, executable trajectories without manual annotation.

---

[1]https://generativelanguage.googleapis.com/v1beta/models/gemini-2.5-flash:generateContent

For inference-time usage, these trajectories are aligned with the task description and used as exemplars; for training-time usage, they are aggregated into a large corpus for supervised fine-tuning.

### 3.3 APPLICATIONS OF TRAJECTORIES

The trajectories extracted from videos can be used in two complementary ways: as in-context exemplars that guide models at inference time, and as supervised data that improve models via fine-tuning.

#### 3.3.1 IN-CONTEXT LEARNING

For in-context learning (ICL), we transform each trajectory into a demonstration that can be inserted directly into a model's context window. Each trajectory consists of *(observation, action)* pairs, but simply showing raw frames and actions may not provide sufficient signal. To improve performance, we prompt Gemini 2.5 Flash to generate natural language rationales for each action in the trajectory, yielding demonstrations of the form $(observation, action, reasoning)$. We format a small set of such demonstrations (typically 3–5) into the input prompt of a general-purpose agent model. At inference time, the agent is conditioned on these exemplars when predicting the next action for a new task, allowing it to draw on planning and grounding priors as well as application-specific knowledge distilled from real demonstrations, without additional training.

#### 3.3.2 SUPERVISED FINE-TUNING

For supervised fine-tuning (SFT), we aggregate the automatically labeled trajectories into a large-scale training corpus. Each trajectory is represented as a sequence of *(state,action)* pairs and used to optimize a multimodal large language model with a standard sequence modeling objective. We train two distinct model families. First, we fine-tune UI-TARS-1.5 (Qin et al., 2025), a strong, open source vision-language-action model designed specifically for computer use. This setting tests whether our trajectories can improve a model that already incorporates domain-specific priors. Second, we fine-tune Qwen 2.5-VL (Bai et al., 2025), a state-of-the-art open-weight multimodal LLM. This setting evaluates whether our data can also benefit general-purpose multimodal models that are not tailored to computer use. Overall, these experiments demonstrate our data's value as a versatile supervision signal, capable of enhancing both specialized CUAs and large, open-source MLLMs.

## 4 EXPERIMENTS

### 4.1 SETUP

#### 4.1.1 MODELS

We evaluate three classes of models.

**General-purpose multimodal models.** Gemini 2.5 Flash (Gemini Team, 2025), OpenAI o3 (OpenAI, 2025a), and Claude 4 Sonnet (Anthropic, 2025) are tested in the in-context learning setting.

**Agentic framework.** We use Jedi (Xie et al., 2025), a state-of-the-art vision-only agentic framework for OSWorld. Jedi couples an MLLM planner (OpenAI o3), which outputs natural-language action steps, with the Jedi-7B grounding model, which maps those steps to executable UI actions. We report results both with and without our trajectories provided as in-context exemplars to the agent.

**Open-source models.** We train UI-TARS-1.5-7B (Qin et al., 2025) and Qwen 2.5-VL 7B Bai et al. (2025) with supervised fine-tuning on our 53,125 video-derived trajectories. This dual evaluation highlights that our data improve both specialized CUAs and general-purpose multimodal models.

#### 4.1.2 DATASETS

Our experiments involve three categories of data.

**State-transition corpus.** To train the IDM, we collect approximately 500k transitions from autonomous web interactions and add 132k human-annotated transitions from Mind2Web (Deng et al., 2023), resulting in over 630k $(O_t, a_t, O_{t+1})$ triples.

**Video-derived trajectories.** Once trained, the IDM is applied to retrieved and filtered YouTube tutorials, producing 53,125 high-quality trajectories across 69 applications spanning productivity, programming, design, screen editing, audio production, system utilities, and scientific/data domains. The category distribution of these trajectories is summarized in Table 1.

As a data labeling baseline, we use TongUI (Zhang et al., 2025), which generates action annotations by prompting the UI-TARS-7B agent. Unlike our video-derived trajectories, these labels are often noisy and inaccurate due to reliance on an imperfect web agent, but they serve as a useful point of comparison for evaluating label quality.

**Evaluation benchmark.** We use *OSWorld-Verified* (Xie et al., 2024), the most up-to-date version of OSWorld, as our primary benchmark. It evaluates agents in real desktop and operating system environments across productivity, programming, design, and system utilities. Tasks must be solved under interactive execution with a 50-step limit, stressing agents' ability to plan, ground instructions in dynamic states, and apply domain knowledge across diverse applications. This makes OSWorld-Verified a comprehensive testbed for both in-context learning and supervised fine-tuning.

| Category | # Apps | # Videos |
|---|---|---|
| Productivity | 11 | 8,691 |
| Programming | 12 | 12,829 |
| Design | 9 | 7,948 |
| Screen Editing | 8 | 7,808 |
| Audio Production | 8 | 5,206 |
| System Utilities | 11 | 4,601 |
| Science & Data | 10 | 6,042 |
| **Total** | **69** | **53,125** |

Table 1: Distribution of collected videos across 69 applications in 7 main categories.

### 4.2 RESULTS AND ANALYSIS

Table 2 summarizes our main results on OSWorld across both in-context learning and supervised fine-tuning. We observe consistent improvements across all model categories. For **general-purpose multimodal models** (Gemini 2.5 Flash, OpenAI o3, Claude 4 Sonnet), adding our W&L exemplars improves performance by +1.6 to +3.0 points. This shows that trajectories distilled from web tutorials provide useful domain-specific priors that even strong foundation models can leverage at inference time. For the **Jedi agentic framework**, which couples the o3 planner with Jedi grounding, W&L yields a +2.2 point gain. This demonstrates that our trajectories can complement structured planning pipelines by enriching them with exemplars that support both planning and grounding. For **open-source CUAs**, supervised fine-tuning on our 53k video-derived trajectories yields even larger gains. UI-TARS-7B improves by +3.8 points, while Qwen 2.5-VL sees the largest improvement, from 1.9 to 13.0 (+11.1). This larger jump is expected because Qwen is a general-purpose multimodal model not originally trained for computer use, so it benefits disproportionately from our dataset, which provides task-specific supervision that was previously missing. Overall, these results highlight the value of our dataset as a scalable supervision signal for both specialized CUAs and broader multimodal models.

#### 4.2.1 HOW MUCH DO LABELED TRAJECTORIES HELP IN IN-CONTEXT LEARNING?

We next analyze the contribution of accurate video labeling to in-context learning (ICL). Our framework provides structured action annotations and natural language reasoning for each step. To isolate the effect of each, we compare three variants: (i) consecutive frames only, (ii) frames paired with predicted actions, and (iii) frames with both actions and reasoning generated by Gemini 2.5 Flash.

Ablations on OSWorld (Table 3) show that adding action labels provides a substantial boost over using frames alone, and further gains are achieved when natural language reasoning is included. This pattern holds consistently across all tested models. Figure 3 provides a qualitative example, showing how labeled trajectories impact the original agent's behavior. The improvement demonstrates that labeled trajectories do more than supply visual context; they encode procedural knowledge that helps models improve both planning and grounding for complex workflows.

#### 4.2.2 HOW DOES LABEL ACCURACY IMPACT PERFORMANCE?

Action label accuracy is central to training CUAs: noisy annotations not only fail to help but can actively degrade performance. We first compare our dedicated IDM against Gemini 2.5 Flash and the TongUI labeling pipeline (based on UI-TARS-7B) on the held-out Mind2Web test set (Table 4).

| Category | Base Model | Method | Success Rate (%) |
|---|---|---|---|
| | *In-Context Learning* | | |
| General Models | Gemini 2.5 Flash (Gemini Team, 2025) | Base (w/o video) | 19.0 |
| | | w/ video; IDM: W&L | **22.0 (+3.0)** |
| | OpenAI o3 (OpenAI, 2025a) | Base (w/o video) | 21.8 |
| | | w/ video; Labeling: TongUI | 21.1 (-0.7) |
| | | w/ video; IDM: W&L | **24.3 (+2.5)** |
| | Claude 4 Sonnet (Anthropic, 2025) | Base (w/o video) | 43.9 |
| | | w/ video; IDM: W&L | **45.5 (+1.6)** |
| Agentic Framework | Jedi (Xie et al., 2025) | Base (w/o video) | 50.6 |
| | | w/ video; IDM: W&L | **52.8 (+2.2)** |
| | *Supervised Fine-Tuning* | | |
| Open-Source Models | Qwen 2.5VL 7B (Bai et al., 2025) | Base (No SFT) | 1.9 |
| | | SFT; Labeling: TongUI | 5.4 (+3.5) |
| | | SFT; IDM: W&L | **13.0 (+11.1)** |
| | UI-TARS-7B (Qin et al., 2025) | Base (No SFT) | 27.3 |
| | | SFT; Labeling: TongUI | 23.8 (-3.5) |
| | | SFT; IDM: W&L | **31.1 (+3.8)** |

Table 2: Main results on OSWorld. W&L improves general multimodal models, an agentic framework, and open-source CUAs across both in-context learning and supervised fine-tuning.

| | Gemini 2.5 Flash | OpenAI o3 | Claude 4 Sonnet |
|---|---|---|---|
| Baseline (no exemplars) | 19.0 | 21.8 | 43.9 |
| + Frames | 18.4 | 21.8 | 43.9 |
| + Frames + Actions | 20.1 | 23.0 | 44.4 |
| + Frames + Actions + Reasoning | **22.0** | **24.3** | **45.5** |

Table 3: Ablation study on the effect of action labeling and reasoning in ICL exemplars (OSWorld success rates). Structured trajectories provide consistent gains over raw frames across all models.

Our IDM achieves the strongest results, substantially outperforming both baselines. TongUI offers some gains over Gemini, especially for structured actions such as `scroll` and `click`, but still falls short of our IDM. A remaining limitation is text decoding for `type` actions, where the margin is smaller.

These differences in labeling accuracy directly translate into downstream performance. TongUI, despite sharing our prompt format, relies on noisy labels that hurt both in-context learning and fine-tuning (Table 4). With o3,

| ActionType | Gemini 2.5 Flash | TongUI | W&L IDM |
|---|---|---|---|
| click(x, y) | 69.2% | 72.7% | **94.4%** |
| scroll(scroll_y) | 50.5% | 76.4% | **93.7%** |
| type(text) | 77.2% | 71.8% | **78.5%** |
| wait(500ms) | 92.3% | 94.1% | **97.5%** |
| move(x, y) | 65.8% | 70.3% | **89.2%** |
| **Action Accuracy** | 72.8% | 82.7% | **91.6%** |
| **ActionType Accuracy** | 81.4% | 88.9% | **96.4%** |

Table 4: Comparison of action labeling accuracy on the Mind2Web test set. W&L's IDM outperforms TongUI, achieving the best performance

TongUI exemplars reduce success rates; in model training, they yield only marginal gains for Qwen and even lower UI-TARS performance (Table 2). In contrast, our IDM-derived labels consistently improve performance, underscoring that reliable supervision is key for effective action grounding.

### 4.2.3 WHAT IS THE EFFECT OF RETRIEVAL QUALITY FOR IN-CONTEXT LEARNING?

We further examine the role of retrieval quality by comparing our method against a random retrieval baseline using o3 (Table 5). Interestingly, random retrieval neither improves nor degrades performance relative to the base model. This suggests that, while carefully retrieved exemplars provide useful signal, even randomly selected exemplars do not introduce significant noise. A likely explanation is that the action labels themselves remain highly accurate regardless of retrieval quality, ensuring

| | o3 (base) | o3 + Random | o3 + W&L |
|---|---|---|---|
| ICL | 21.8 | 21.8 | **24.3 (+2.5)** |

Table 5: ICL results on OSWorld with o3. Random retrieval has little effect, while W&L yields strong gains.

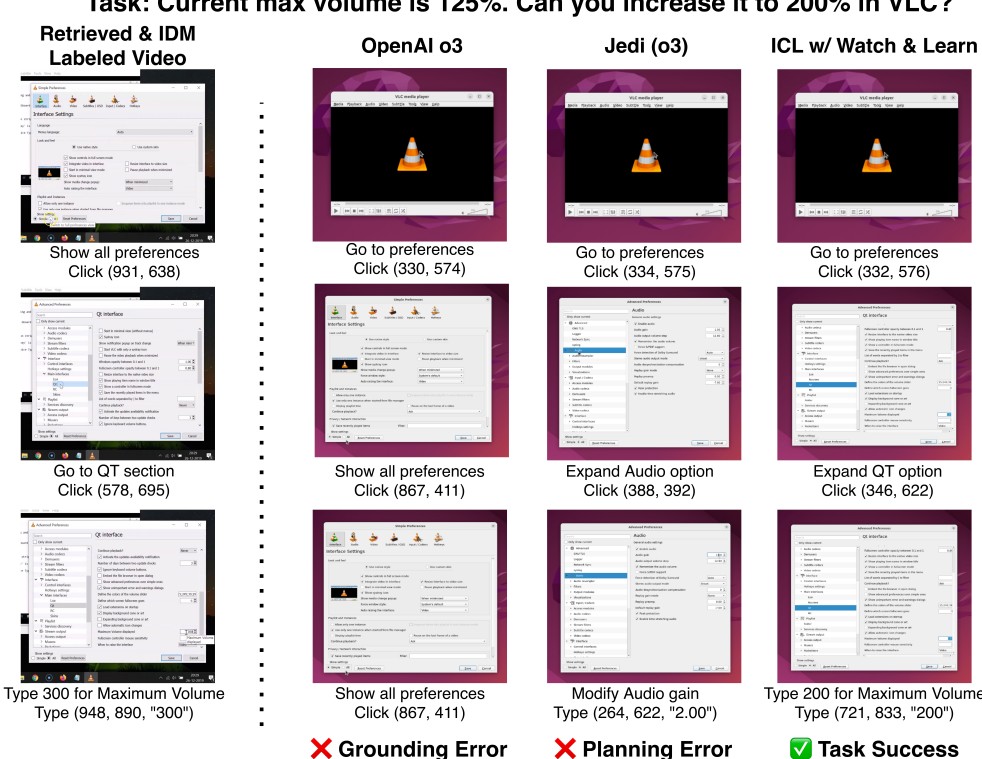

Figure 3: Qualitative examples on OSWorld. On the left, the video-derived trajectory that W&L generates for the task. On the right: (i) the o3 agent makes a grounding error by selecting a wrong UI element; (ii) the Jedi (o3) agent makes a planning error by entering the wrong submenu without recovering; (iii) using the video-derived trajectory, W&L agent completes the task successfully. Images are cropped for visibility, and the action coordinates correspond to the original full-resolution screenshots.

that the model is not misled by contradictory supervision. These results indicate that the main benefit of our method lies in providing *targeted* exemplars that align closely with the task context. Retrieval quality therefore determines the strength of the positive effect, but poor retrieval does not actively harm performance when the underlying labels are still correct.

## 5 CONCLUSION AND FUTURE WORK

We introduced W&L, a framework that transforms web-scale human demonstration videos into executable UI trajectories using a vision-only IDM and a task-aware retrieval pipeline. With over 53k automatically labeled trajectories, we showed improvements in both in-context learning and supervised fine-tuning, benefiting general-purpose MLLMs as well as specialized CUAs.

Our experiments on OSWorld highlight that (i) a dedicated IDM provides stronger action prediction than foundation models, (ii) action-labeled exemplars improve ICL and SFT, (iii) domains with abundant tutorials see larger gains, and (iv) performance scales with more training data and better retrieval.

Looking ahead, we plan to extend the IDM to richer actions such as drag-and-drop, combine or split tutorials to better construct long-horizon trajectories, and explore reinforcement learning with our trajectories—using them as demonstrations for behavior cloning, as replay buffers for offline RL, or as priors for reward modeling in online training. These directions can further bridge large-scale demonstrations with adaptive learning, pushing CUAs closer to real-world deployment.

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
