## APPENDICES

In this supplementary material, we present additional details and clarifications that are omitted in the main text due to space constraints.

- Appendix A Use of Large Language Models (LLMs).
- Appendix B Limitations.
- Appendix C Dataset Details.
- Appendix D Implementation Details.
- Appendix E More Results.

## A  USE OF LARGE LANGUAGE MODELS (LLMS)

Large language models (LLMs) were used in limited ways during this work. Specifically, we used LLM-based assistants to (i) improve sentence structure, paragraph organization, and grammar in the writing process, and (ii) provide coding assistance such as debugging and suggesting alternative implementations. LLMs were not used for research ideation, experimental design, or analysis. All scientific contributions, including problem formulation, methodology, experiments, and conclusions, are solely the work of the authors.

## B  LIMITATIONS

While our framework demonstrates strong performance, there remain several opportunities for extension. First, our inverse dynamics model (IDM) currently focuses on a core set of primitive actions such as click, type, move, and scroll. More complex actions like drag-and-drop are not yet supported, largely due to the absence of sufficient training data. Similarly, while our IDM predicts scroll actions, we were unable to curate a large-scale, diverse dataset of scrolling behaviors from web interactions, which limits its robustness in this dimension. Expanding the action space with richer interaction types and collecting more representative scroll data are promising directions.

Second, our retrieval framework retrieves demonstrations at the granularity of full tasks. While effective, this may not always align with the granularity needed by an agent during execution. Future work could explore mechanisms to automatically merge shorter tasks into longer workflows, or split lengthy tutorials into more targeted sub-trajectories. Such advances would enable finer-grained retrieval and more flexible trajectory construction, ultimately improving the adaptability of our approach.

We view these limitations not as fundamental barriers but as natural opportunities to further enhance the scalability and generality of our framework.

## C  DATASET DETAILS

### C.1  APPLICATIONS BY CATEGORY

We selected seven categories: **Productivity, Programming, Design, Screen Editing, Audio Production, System Utilities, and Science & Data**. These categories span a broad range of realistic computer use. Productivity tools (e.g., Microsoft Office, Google Workspace) cover everyday document and collaboration tasks, while Programming environments (e.g., VS Code, Jupyter) capture software development workflows. Design (e.g., Photoshop, Figma), Screen Editing (e.g., Premiere Pro, OBS Studio), and Audio Production (e.g., Audacity, FL Studio) extend to creative domains with specialized interfaces. System Utilities (e.g., Task Manager, Finder, Docker) test low-level system interaction, and Science & Data tools (e.g., MATLAB, Tableau, SPSS) represent analytical and visualization tasks.

Applications within each category were chosen for their widespread adoption, abundant tutorial availability on YouTube, and ability to showcase the diverse interaction challenges agents must master. While we focused on these applications, our method is not restricted to them: additional data

| Category | Applications |
|---|---|
| Productivity | Microsoft Office, Google Workspace, Notion, Evernote, OneNote, Trello, Asana, ClickUp, Monday.com, Slack, Microsoft Teams |
| Programming | VS Code, PyCharm, IntelliJ IDEA, Eclipse, Android Studio, Xcode, Jupyter Notebook, Google Colab, RStudio, Sublime Text, Atom, GitHub Desktop |
| Design | Adobe Photoshop, Adobe Illustrator, Adobe XD, Figma, Sketch, Canva, CorelDRAW, Inkscape, Affinity Designer |
| Screen Editing | Adobe Premiere Pro, Final Cut Pro, DaVinci Resolve, Camtasia, OBS Studio, ScreenFlow, Filmora, iMovie |
| Audio Production | Audacity, Adobe Audition, FL Studio, Logic Pro X, Ableton Live, Pro Tools, Cubase, GarageBand |
| System Utilities | Windows Task Manager, PowerShell, macOS Finder, Activity Monitor, Disk Utility, Linux Terminal, Docker, VirtualBox, CCleaner, WinRAR, 7-Zip |
| Science & Data | MATLAB, Mathematica, SPSS, SAS, Tableau, Power BI, Google Colab, Jupyter Notebook, Stata, RapidMiner |

Table 6: Applications grouped by category.

can be generated from any new tutorial videos available on the web. The distribution of applications is in Table 6.

## D    IMPLEMENTATION DETAILS

### D.1    VIDEO RETRIEVAL

To build a large-scale dataset of application demonstrations, we require a method to identify relevant tutorial videos from the web. YouTube is a natural source since it contains abundant tutorials across productivity, programming, design, and other domains. However, naively searching by task description may yield irrelevant or entertainment-focused videos. To address this, we designed a dedicated prompt for generating targeted search queries.

The prompt (shown below) instructs a language model to act as an expert in YouTube search, taking as input a task description and a list of related applications. It outputs a short and effective query that emphasizes tutorials, how-to videos, and instructional content. By constraining queries to be concise and domain-specific, this approach improves retrieval precision and reduces noise from unrelated videos.

---

**Prompt for Video Retreival Query Generation**

You are an expert at creating YouTube search queries. Given a task instruction and related applications, create a concise, effective search query that will find relevant tutorial videos.

Task: {instruction}
Related Applications: {related_apps}

Create a search query that would find helpful tutorial videos for this task. Focus on tutorial, how-to, or instructional content. Keep it concise (under 10 words).
Search query:

---

### D.2    VIDEO FILTERING

After retrieving candidate tutorials, many videos still contain irrelevant or low-quality content such as talking-head introductions, presentation slides, or animated transitions. To ensure that our dataset is composed of high-quality screen recordings that clearly demonstrate application use, we apply a filtering step.

We design a prompt that instructs a language model to act as a visual classifier. Given a single frame from a video, the model assigns both a categorical label (e.g., clean screencast, zoomed screencast, talking head) and a quality score between 0.0 and 1.0. We retain only those videos where the average frame score exceeds 0.8, which empirically yields a reliable set of clean tutorial screencasts. This threshold balances recall and precision: it removes noisy or non-screencast content while retaining a broad coverage of genuine tutorials.

---

**Prompt for Video Filtering**

You are a visual classifier helping to filter video tutorial frames for clean screencast content.
Your task is to classify an input image (a single frame from a video) and provide a quality score.

Classify the image into one of these categories:
1. Clean Screencast: Full desktop screen showing software interface, application window, code editor, browser, or terminal. Clear, unzoomed view of the entire screen or application window.

2. Zoomed Screencast: Screenshot that has been zoomed in or cropped, showing only part of the screen or interface elements.

3. Animated/Transition: Frames with animations, transitions, intro/outro effects, or visual effects that are not static screencast content.

4. Talking Head: Person's face or upper body from webcam, typically in corner or overlay.

5. Slide/Presentation: Static presentation slide, diagram, or text-heavy content.

6. Other: Content that doesn't fit the above categories.

For each classification, also provide a quality score from 0.0 to 1.0: - 1.0: Perfect clean screencast - 0.8-0.9: Good screencast with minor issues - 0.6-0.7: Acceptable screencast - 0.4-0.5: Poor quality or partially zoomed - 0.0-0.3: Very poor or not screencast

Return your response in this format: Category: [category name] Quality: [score] Reason: [brief explanation]

---

### D.3 MODELS

For in-context learning evaluations we query API-based models using their latest public versions: Google Gemini 2.5 Flash (`gemini-2.5-flash`), OpenAI o3 (`o3-2025-04-16`), and Anthropic Claude 4 Sonnet (`claude-4-sonnet-20250514`). We use deterministic decoding with temperature set to 0.0.

For IDM training, we use the AdamW optimizer with a learning rate of $3e-4$, batch size 256, and cosine learning rate decay. Training is run for 15 epochs on $8\times$A100 GPUs (80GB) with gradient clipping at 1.0 and mixed-precision (bfloat16). For supervised fine-tuning of CUAs, we follow the official training recipes from UI-TARS-1.5 and Qwen 2.5-VL, adapting batch size to fit the same hardware setup.

## E MORE RESULTS

### E.1 WHAT IS THE EFFECT OF DATA SCALE FOR SUPERVISED FINE-TUNING?

| Model | Base | 10k | 25k | Full |
|---|---|---|---|---|
| Qwen 2.5-VL | 1.9 | 3.3 | 4.9 | 13.0 |

Table 7: Data scaling results on OSWorld with Qwen 2.5-VL. Performance improves as training data increases from 10k to 25k and the full dataset.

We study how scaling the number of training trajectories affects the performance of Qwen 2.5-VL on OSWorld. As shown in Table 7, success rates increase from 1.9% with the base model to 3.3% with 10k trajectories, 4.9% with 25k trajectories, and 13.0% with the full dataset. The improvement is closer to exponential than linear, suggesting that a minimum critical mass of data is required before substantial gains emerge.

We hypothesize that this behavior arises because Qwen must learn both *grounding* and *planning* from the video-derived trajectories. With limited data, the model struggles to acquire either capability robustly, leading to only small improvements. Once enough trajectories are available, however, Qwen begins to effectively integrate grounding of UI states with coherent planning patterns, producing sharper gains. This indicates that further scaling of high-quality trajectories could unlock even larger benefits.

## E.2 WHICH APPLICATION DOMAINS BENEFIT MOST FROM OUR DATA?

| Setting | Category | Model | chrome | gimp | lo_calc | lo_impress | lo_writer | multi_apps | os | thunderbird | vlc | vs_code | Total |
|---|---|---|---|---|---|---|---|---|---|---|---|---|---|
| ICL | General Models | Gemini 2.5 Flash | 8 | 8 | 4 | 3 | 5 | 9 | 10 | 6 | 5 | 12 | 70 |
| | | + W&L | **10 (+3)** | **10 (+2)** | 4 | 5 | 5 | 9 | 10 | **8 (+2)** | **8 (+3)** | 12 | **81 (+11)** |
| | | o3 | 6 | 10 | 5 | 5 | 7 | 15 | 15 | 4 | 7 | 9 | 83 |
| | | + W&L | **9 (+3)** | **13 (+2)** | **7 (+1)** | 7 | 7 | **18 (+1)** | 15 | 4 | **9 (+2)** | 9 | **98 (+9)** |
| | | Claude 4 Sonnet | 25 | 13 | 15 | 22 | 14 | 27 | 11 | 11 | 7 | 14 | 159 |
| | | + W&L | **27 (+2)** | **15 (+2)** | 15 | 22 | 14 | 27 | 11 | 11 | **9 (+2)** | 14 | **169 (+6)** |
| | Agentic Framework | Jedi | 26 | 21 | 19 | 21 | 15 | 32 | 13 | 12 | 10 | 13 | 182 |
| | | + W&L | **29 (+3)** | **23 (+2)** | 19 | **23 (+2)** | 15 | 32 | 13 | 12 | **12 (+2)** | 13 | **191 (+9)** |
| SFT | Open-Weight Models | UI-TARS-7B | 11 | 15 | 6 | 14 | 9 | 5 | 8 | 4 | 6 | 15 | 93 |
| | | + W&L | **13 (+2)** | **17 (+2)** | **8 (+2)** | **16 (+2)** | 9 | **7 (+2)** | 8 | **4 (+2)** | **7 (+2)** | 15 | **104 (+14)** |
| | | Qwen 2.5-VL 7B | 4 | 1 | 0 | 0 | 2 | 0 | 0 | 2 | 2 | 0 | 7 |
| | | + W&L | **12 (+8)** | **10 (+9)** | **3 (+3)** | **1 (+1)** | 2 | **1 (+1)** | **5 (+5)** | **4 (+2)** | **6 (+4)** | **4 (+4)** | **48 (+41)** |

Table 8: Detailed OSWorld category-wise task successes. W&L provides the strongest improvements in domains with abundant specialized tutorials (e.g., Chrome, Gimp, VLC), while gains are smaller in domains requiring heavy text entry, rare actions, or fine-grained control.

To better understand the strengths and limitations of our approach, we break down results by application domain on OSWorld. Table 8 reports task successes for general-purpose models (o3, Claude 4 Sonnet), the Jedi agentic framework, and the open-source model UI-TARS-7B, both with and without W&L exemplars or training data.

The largest improvements are observed in `chrome`, `gimp`, and `vlc`. These domains benefit strongly from specialized procedural knowledge that is well covered by online tutorials, such as configuring browser settings, editing images, or adjusting media player preferences. The presence of abundant, step-by-step demonstrations in these categories enables our pipeline to extract high-quality trajectories that transfer effectively to downstream agents.

By contrast, the gains are smaller in domains such as `vscode` and `os`, which often require extensive text entry or code manipulation—capabilities that are less easily captured by our current action set. Improvements are also limited in `thunderbird` and LibreOffice applications (`lo.calc`, `lo.writer`, `lo.impress`), where high-quality tutorials are scarce and tasks sometimes involve fine-grained interactions such as dragging objects or manipulating small interface elements. These are challenging for our IDM that does not yet support drag-and-drop actions.

Overall, this breakdown highlights a key property of our approach: it yields the largest benefits in domains where web tutorials are both plentiful and aligned with the action space of the agent, while leaving room for future extensions in text-heavy or fine-grained interaction domains.