# OpenReview forum: "Watch and Learn: Learning to Use Computers from Online Videos"
_ICLR.cc/2026/Conference — ICLR 2026 Conference Withdrawn Submission_

### Official Review · Reviewer_SyFe · 2025-10-31

**Soundness:** 3
**Presentation:** 2
**Contribution:** 2
**Rating:** 4
**Confidence:** 2

**Summary:**

The paper proposes Watch & Learn, a pipeline that mines tutorial/screencast videos and converts them into executable GUI trajectories for training and prompting computer-use agents. Instead of heuristic labelers, the authors train a vision-only inverse dynamics model that predicts the user action between successive frames and its arguments. The IDM is trained on a large mix of synthetic and real transitions, then applied to retrieved/filtered videos to produce a sizable corpus of trajectories covering many desktop/web applications. These mined trajectories are used (i) as in-context demonstrations and (ii) for supervised fine-tuning of open CUAs, yielding notable gains on OSWorld-Verified and improvements when plugged into agent frameworks like Jedi.

**Strengths:**

- Originality & framing. Casting GUI action recovery as inverse dynamics is clean and avoids brittle multi-stage MLLM pipelines.
- Clarity & design. The IDM components (visual encoder, transformer heads for action type/coords/text) and multi-task objectives are described clearly.
- Empirical quality. Better action labeling than prior labelers translates to downstream improvements for both ICL and SFT on a standard benchmark.

**Weaknesses:**

- **Typed-text accuracy is a bottleneck.** The method improves clicks/scrolls/moves, but the margin on `type(text)` is modest; the paper does not show targeted strategies for language input disambiguation; the authors should add OCR-aided candidate extraction, focus-aware lexicons, and constrained decoding with CER/WER metrics.
- **Limited action space.** Only five primitives are supported; drag-and-drop, hotkeys, hover, resize, multi-window are out of scope. The paper does not quantify failures from these omissions; the authors should extend the action set and report a breakdown of tasks requiring richer actions.
- **Temporal granularity.** Fixed low FPS risks action-boundary drift. The paper does not show sensitivity to sampling rate; the authors should compare 1/2/4+ FPS and include alignment metrics on a timestamped subset.
- **Coverage vs. benchmark overlap.** The paper does not analyze domain-shift between mined tutorials and OSWorld tasks (app/version/locale/resolution). The authors should report per-domain coverage and success by app/version.
- **Causal attribution.** Label-quality → downstream gains are suggested, but not isolated. The paper does not show a controlled swap that holds retrieval/rationales fixed while varying only labels; the authors should include such a study.

**Questions:**

1. What are CER/WER and edit-distance stats for `type(text)`? Do OCR-assisted candidate sets improve robustness?
2. Why fix 1 fps? Please provide an fps vs. labeling-F1 vs. OSWorld-success sensitivity curve; does adaptive sampling around rapid UI changes help?
3. What fraction of failures stem from unsupported actions (drag-drop, hotkeys, hover)? Any preliminary results with an extended action set?
4. Can the retrieval/filtering/rationale steps be replicated with open VLMs (e.g., Qwen2.5-VL/LLaVA-Next) with a report on cost/quality trade-offs?
5. Any results for other OSes (Win/Linux/macOS variants) or datasets (e.g., Android/web automation suites)?

I am still confusing about the so-called "53k high-quality trajectories from raw web video". How to define the high-quality since the trajs are  labelled by inverse dynamics pipeline. And how does the false-negative and false-positive data affect?

---

### Official Review · Reviewer_oFYi · 2025-10-31

**Soundness:** 3
**Presentation:** 3
**Contribution:** 3
**Rating:** 6
**Confidence:** 3

**Summary:**

This paper presents "Watch & Learn" (W&L), a scalable framework that converts raw, web-crawled tutorial videos into executable UI trajectories for training computer-use agents (CUAs). The core technical idea is to treat action extraction as an inverse-dynamics problem: given two consecutive screenshots, a specialized vision model (IDM) predicts the intervening action. An automatic pipeline retrieves relevant YouTube videos, filters non-informative segments, applies the IDM, and outputs 53k labeled trajectories covering 69 applications. The trajectories are used in two ways: (i) as in-context exemplars at inference time, and (ii) as supervision for fine-tuning open-source models. On the OSWorld benchmark, W&L improves strong closed models (Gemini 2.5 Flash, o3, Claude 4) by +1.6–3.0 pp in-context, and boosts open-weight models (Qwen 2.5-VL, UI-TARS) by up to +11.1 pp after SFT, without any human annotation.

**Strengths:**

Novel perspective: First work to adapt inverse-dynamics learning (common in robotics) to GUI action extraction, sidestepping brittle multi-stage MLLM pipelines.
Large-scale, high-quality data: 53 k trajectories with 91.6 % action-label accuracy (vs. 72.8 % for Gemini 2.5 Flash and 82.7 % for TongUI), enabling reproducible progress.
Dual utility: Demonstrates that the same trajectories help both in-context (test-time) and fine-tuning (training-time), a rare combination in prior CUA work.

**Weaknesses:**

See questions.

**Questions:**

The paper introduces a novel approach to collecting data in computer-use agents by adapting inverse-dynamics learning from robotics to GUI domains. The subsequent deployment of these automatically mined trajectories for both in-context learning and fine-tuning, with gains of up to 11.1% on OSWorld.
Nevertheless, two issues deserve clarification:
1. Performance drop with TongUI labels. Table 2 shows that UI-TARS-7B fine-tuned on TongUI-produced trajectories drops from 27.3% to 23.8%, whereas training with W&L labels rises to 31.1%. Is this reversal solely attributable to label noise, or could it reflect a hyper-parameter mismatch (e.g., learning rate) that inadvertently overfits to TongUI’s erroneous actions?
2. Ablation on IDM training data composition. The IDM is trained on 500k self-collected synthetic transitions plus 132k Mind2Web human annotations. Without an ablation that ablates either subset, it is unclear whether the proposed automatic web-crawl interaction pipeline actually moves the needle. Could the authors report IDM accuracy when trained only on Mind2Web (132k) versus only on the new crawl (500k) versus the union? Such a table would validate that the synthetic collection strategy is not just scalable but also necessary for the strong downstream numbers presented.

---

### Official Review · Reviewer_EuS1 · 2025-11-03

**Soundness:** 2
**Presentation:** 2
**Contribution:** 1
**Rating:** 2
**Confidence:** 3

**Summary:**

This paper presents a approach to solve computer tasks by querying related videos from youtube, label the trajectories with an inverse dynamics model, and update the agent with SFT on the labeled trajectories. The paper reports 3-5% improvements in computer use task performance after updating with their approach. They also report 2-3% improvements when used as an in-context learning approach with closed models.

**Strengths:**

- Interesting approach that presents itself as amenable to both SFT and ICL.
- Decent evaluation of agents across various backbone.

**Weaknesses:**

- Lack of examples. The reviewer only noted a single example (Figure 3) of the agent's web interactions. More examples would convince the reviewer that the agent is indeed functional.
- No measure of IDM performance. Training an IDM for this task is not trivial. More explanation as to how it is trained, success rates, and more empirical evidence of its performance should be included.

**Questions:**

Can you add more examples of the agent being used?

How does the IDM perform on benchmarks and eval scenarios?

How are in context examples formatted?

---

### Official Review · Reviewer_cfK8 · 2025-11-04

**Soundness:** 3
**Presentation:** 3
**Contribution:** 3
**Rating:** 4
**Confidence:** 4

**Summary:**

This work introduces Watch & Learn (W&L), a framework that leverages web-scale human demonstration videos to generate executable UI trajectories. These trajectories can be used both for ICL and as SFT data for CUAs. The approach employs an IDM to predict user actions from consecutive screen states, resulting in high-quality trajectories across diverse applications. The authors demonstrate that their method significantly enhances performance on benchmarks like OSWorld, outperforming general-purpose models and specialized CUAs.

**Strengths:**

1. The primary contribution is a scalable, annotation-free method for trajectory generation. Using an IDM to infer actions from state transitions is an elegant approach that avoids reliance on brittle multi-stage heuristics or costly online exploration.
2. The W&L-generated trajectories show immense value as SFT data. For general-purpose models like Qwen 2.5-VL, the performance improvement is a substantial percentage point (11.1).

**Weaknesses:**

1. Although the ICL scenario is very interesting, the improvements are relatively modest. In OS-World, it only resulted in a slight improvement, such as correctly predicting half or one more task for +3% accuracy.
2. As shown in Table 4, while the overall IDM accuracy is high, its performance on the type(text) action (78.5%) is notably lower than for actions like click (94.4%) or wait (97.5%). This could limit the framework's effectiveness on tasks requiring complex text entry.

**Questions:**

See Weaknesses.

---

### Author Response · Authors · 2025-11-14

We thank the reviewers for the time and effort they invested in evaluating our submission. We have decided to withdraw the paper because the new material we plan to add will significantly change the scope and content of the work. Before withdrawing, we would like to clarify several points where misunderstandings may have arisen.

### **Reviewer cfK8**

**Comment: OSWorld improvements are minor.**
The average successful OSWorld task takes about nineteen steps. An incorrect prediction at any step results in a full failure, so even a small increase in success rate is difficult to achieve. A three percent improvement corresponds to roughly nine additional successful tasks out of 355, which reflects a meaningful gain.

**Comment: Text accuracy of the IDM is lower.**
We recognize this limitation. After the initial submission, we improved the text accuracy to 86% (from ~78%) using data augmentation with 100k additional state transition samples.

### **Reviewer EuS1**

**Comment: Lack of examples.**
We apologize for any doubt caused by the limited number of examples. A future version will include more qualitative illustrations. The reported results are genuine and supported by extensive experiments.

**Comment: No measure of IDM performance.**
Table 4 reports the accuracy of our IDM on the Mind2Web test split, exceeding 91% and outperforming TongUI at 82.7%.

### **Reviewer oFYi**

**Comment: Can the performance drop with TongUI be attributed solely to label accuracy.**
We controlled for all other factors and kept the training setup the same across comparisons, varying only the label source to ensure a fair evaluation.

**Comment: IDM training data composition.**
In later experiments, we achieved higher accuracy after removing Mind2Web data. Despite being human annotated, Mind2Web contains a noticeable amount of noise, whereas our state transition labeling pipeline collects only valid transitions.

### **Reviewer SyFe**

**Comment: Text accuracy bottleneck.**
As noted above, we improved text accuracy to 86% through data augmentation.

**Comment: Why one FPS.**
We selected 1 FPS based on empirical observation. YouTube tutorial videos tend to progress slowly and contain few rapid actions. We tested 2 and 5 FPS and did not observe meaningful downstream degradation / improvement. The choice remains a tunable hyperparameter.

**Comment: Do unsupported actions lead to failure.**
We extended the action set to include drag and drop by modeling it as click down, move, and release, and to include hotkeys using the same mechanism as text entry. Hover is already covered by the move action. Existing agents, aside from Qwen2.5VL-7B, can still use their full native action sets, since our trajectories do not constrain their available actions.

**Comment: Can the retrieval, filtering, and rationale steps be replicated with open VLMs.**
We have not conducted a broad study with open models. Initial experiments showed difficulty in filtering out non screencast videos and detecting zoom variations.

**Comment: Any results on other operating systems.**
We ran experiments on WindowsAgentArena using a fine tuned UI-TARS-1.5-7B model and obtained strong performance. Our retrieval pipeline naturally gathers videos from diverse operating systems, with YouTube content skewed toward Windows and macOS. This distribution aligns with our observations. We did not retrieve mobile videos because they require a different action set such as pinch or rotate.

---

### Note · Authors · 2025-11-14

I have read and agree with the venue's withdrawal policy on behalf of myself and my co-authors.